# Is amalgam removal in patients with medically unexplained physical symptoms cost-effective? A prospective cohort and decision modelling study in Norway

**Admassu N. Lamu**[1,2]*, **Lars Björkman**[3,4], **Harald J. Hamre**[5], **Terje Alræk**[6], **Frauke Musial**[6], **Bjarne Robberstad**[2]

**1** Department of Community Medicine, UiT The Arctic University of Norway, Tromsø, Norway, **2** Section for Ethics and Health Economics, Department of Global Health and Primary Care, University of Bergen, Bergen, Norway, **3** Dental Biomaterials Adverse Reaction Unit, NORCE Norwegian Research Centre AS, Bergen, Norway, **4** Department of Clinical Dentistry, University of Bergen, Bergen, Norway, **5** Institute for Applied Epistemology and Medical Methodology, University of Witten/Herdecke, Witten, Germany, **6** National Research Center in Complementary and Alternative Medicine, NAFKAM, Department of Community Medicine, UiT The Arctic University of Norway, Tromsø, Norway

* kanaroba2.12@gmail.com

**Data Availability Statement:** The data contain potentially identifying and sensitive patient information and is not available due to personal

## Abstract

There are many patients in general practice with health complaints that cannot be medically explained. Some of these patients attribute their health complaints to dental amalgam restorations. This study examined the cost-effectiveness of the removal of amalgam restorations in patients with medically unexplained physical symptoms (MUPS) attributed to amalgam fillings compared to usual care, based on a prospective cohort study in Norway. Costs were determined using a micro-costing approach at the individual level. Health outcomes were documented at baseline and approximately two years later for both the intervention and the usual care using EQ-5D-5L. Quality adjusted life year (QALY) was used as a main outcome measure. A decision analytical model was developed to estimate the incremental cost-effectiveness of the intervention. Both probabilistic and one-way sensitivity analyses were conducted to assess the impact of uncertainty in costs and effectiveness. In patients who attribute health complaints to dental amalgam restorations and fulfil the inclusion and exclusion criteria, amalgam removal is associated with modest increase in costs at societal level as well as improved health outcomes. In the base-case analysis, the mean incremental cost per patient in the amalgam group was NOK 19 416 compared to the MUPS group, while mean incremental QALY was 0.119 with a time horizon of two years. Thus, the incremental costs per QALY of the intervention was NOK 162 680, which is usually considered cost effective in Norway. The estimated incremental cost per QALY decreased with increasing time horizon, and amalgam removal was found to be cost saving over both 5 and 10 years. This study provides insight into the costs and health outcomes associated with the removal of amalgam restorations in patients who attribute health complaints to dental amalgam fillings, which are appropriate instruments to inform health care priorities.

data protection regulations. The restrictions are imposed by NORCE's Administrative Support for Research and the Data Protection Officer at NORCE. Data requests may be sent to NORCE's Administrative Support for Research (forskningsstotte@norceresearch.no) with reference to "Project 42564 Prospektiv kohortstudie av helseplager."

**Funding:** LB received the funding. The project was funded by Norwegian Ministry of Health and Care Services (https://www.regjeringen.no/no/dep/hod/id421/) via the Norwegian Directorate of Health (Grant no.: NA). The funders had no role in study design, data collection and analysis, decision to publish, or preparation of the manuscript.

**Competing interests:** The authors have declared that no competing interests exist.

## Introduction

A relatively high number of patients in general practice report health complaints that cannot be medically explained. The prevalence of medically unexplained physical symptoms (MUPS) generally varies ranging between 10 and 33% of consultations in primary care [1,2]. In Norwegian general practice, the prevalence of consultation rates of patients with *persistent* MUPS is 3% [2]. Persistent MUPS is defined as physical symptoms with no identified organic cause, lasting for at least three months and leading to a loss of function (explained as absence due to illness or disability, or withdrawal from social activities like sports, social events, or other leisure activities). Some of these patients attribute their health issues to dental amalgam restorations. Dental amalgam was traditionally considered a safe, affordable and durable material, which is used to restore teeth for over a century and a half [3,4]. However, concerns over the release of mercury from amalgam fillings have grown steadily over the past decades. Estimates of the daily dose of mercury vapour from amalgam fillings indicate that levels that can be defined as tolerable can be exceeded [5,6]. Furthermore, autopsy studies have shown that dental amalgam is the main source of inorganic mercury in human tissues, and responsible for a considerable proportion of deposits of inorganic mercury [7]. In some studies, dental amalgam fillings have been linked to neurological diseases, such as Parkinson and Alzheimer diseases [8,9]. Nonetheless, the evidence for such a link is weak. In addition to health-related issues, there is a general concern about environmental effects from mercury.

In response to these concerns, most countries are planning to phase-out the use of amalgam for dental restoration or have already done so [10]. With advancement in technology and materials, there is an increasing tendency to use mercury-free and environmental-friendly materials and avoid the use of dental amalgam [11]. Although there is no strong evidence that amalgam restorations can cause general health symptoms, there are several studies showing that general health complaints are significantly reduced after amalgam removal [12–16]. Using similar data, a recent study showed a considerable reduction of general health complaints after amalgam removal in patients with attribution to their health complaints to amalgam restorations [17]. This effect could be mediated both via specific (reduced exposure to dental amalgam and mercury) and non-specific treatment effects (e.g., expectations, general care, placebo). However, due to design limitations of the available studies, especially the difficulty of conducting blinded studies, it is difficult to draw firm conclusions about a causal relationship between amalgam removal and symptom reduction. Although the use of dental amalgam has been declining over the years, amalgam is still considered a treatment option in several countries [18,19], but phased-out in Norway [20].

As a response to the public expectations, the Norwegian Ministry of Health and Care Services initiated the present project aiming at patients with health complaints attributed to dental amalgam fillings [16]. Since amalgam removal involves additional treatment costs, reliable and evidence-based information on the long-term costs and outcomes of an intervention is needed in order to support policy makers in making well informed decisions. Cost-effectiveness analyses are appropriate instruments to inform about health care priorities and to underpin the development of treatment guidelines [21].

Whilst economic evaluations in dentistry have generally been applied to study alternative forms of dental restorations [22–25], this is, to our knowledge, the first study aiming to perform a classic economic evaluation of dental amalgam filling removal for general health reasons. The current analysis estimates the cost-effectiveness of amalgam removal in patients with MUPS attributed to amalgam fillings (Amalgam cohort) compared to usual care, defined as patients with MUPS without attribution to amalgam fillings and no removal of amalgam fillings (MUPS cohort). The analysis is based on evidence from a prospective cohort study in

Norway. As the intervention has cost implications for both patients and the health services, the analysis is performed from the perspective of society.

## Methods

### Study design

The design of the study was a prospective cohort with non-equivalent comparison group and with pre- and post-test [16]. Participants aged 20 to 70 years were recruited between March 2013 and December 2015. Three groups were recruited separately: i) the main target group consisted of patients with MUPS, which they attributed to dental amalgam restorations (Amalgam cohort); ii) patients with MUPS without attribution to amalgam fillings recruited from general practice (MUPS cohort); and iii) a healthy group primarily recruited at dental practices (Healthy cohort). All respondents should be permanent residents of Norway and able to comply with the protocol. The presence of at least one amalgam filling was the primary criterion for inclusion in Amalgam cohort. Other main inclusion criteria for the Amalgam cohort were unspecific health complaints attributed to dental amalgam restorations at least for three months, wish to have all amalgam fillings removed, and absence of major complication risks following amalgam removal. No attribution to amalgam and no explicit wish to remove amalgam were the key criteria in the MUPS cohort. In both Amalgam and MUPS cohorts, any diagnosed disease should be adequately treated to be included. Detailed recruitment and eligibility criteria as well as power calculation were reported elsewhere [16].

This economic evaluation is based on the Amalgam cohort (intervention group) and MUPS cohort (comparator group) and included participants who responded to both the baseline and follow-up questionnaires. Because the study did not involve an equivalent control group, the comparator group included patients with MUPS who did not relate their symptoms to the presence of amalgam fillings, and who had a similar symptom load as the Amalgam cohort. Data for the healthy cohort were not utilised, as this group was not considered eligible for the intervention. Baseline sample characteristics are presented in Table 1.

Follow-up in the Amalgam cohort was 12 months after removal of the last amalgam filling. The guidelines from the Norwegian Directorate of Health regarding examination and treatment of patients with side effects from dental restorative materials recommend that amalgam restorations should be done gradually without too many restorations in the same session [26]. Thus, with several restorations the procedure could take a year. The mean number of amalgam surfaces for the amalgam cohort was 20.3 (SD: 10.9; range: 5 to 59), and the concentrations of inorganic mercury in serum (a biomarker of exposure) were similar in the two cohorts (p = 0.887; Table 1). Amalgam fillings were, in most cases, replaced with polymer based composite materials, but when indicated large fillings were replaced with metallo-ceramic or ceramic crowns. Mean length of the treatment period (from acceptance to start treatment to finished treatment) was 253 days (SD 134, range from 29 to 619). The more amalgam surfaces to replace, the longer the treatment period was. Thus, since amalgam removal was estimated to take an average of about 12 months, the follow up of the MUPS cohort was 24 months after baseline.

### Setting and perspective

This study adopted a societal perspective in the context of the Norwegian health care setting, and included costs carried both by patients and the health services. In Norway, dental restorations are generally paid out-of-pocket by adult patients, but if incorporated into the public health benefit package, most of the cost would be covered by public budgets. The cost-effectiveness analysis was performed over *two years* as base case analysis, and over five- and ten-year periods in scenario analyses.

**Table 1. Baseline characteristics.**

| Variable (n, %) | Amalgam (n = 32) | | MUPS (n = 28) | |
|---|---|---|---|---|
| Age (Mean, SD) | 52 | 7.5 | 50 | 10.3 |
| Gender | | | | |
| Female | 19 | 59.4 | 24 | 85.7 |
| Male | 13 | 40.6 | 4 | 14.3 |
| Education | | | | |
| Lower and upper secondary | 14 | 43.8 | 17 | 60.7 |
| College, <4yrs | 11 | 34.4 | 9 | 32.1 |
| College, 4+ years | 7 | 21.9 | 2 | 7.1 |
| Live without partner (Yes/No) | | | | |
| Yes | 26 | 81.3 | 23 | 82.1 |
| No | 16 | 18.7 | 5 | 17.9 |
| Income (in '000) | | | | |
| Low, < 450 | 8 | 25.8 | 4 | 14.3 |
| Middle, 450–750 | 13 | 41.9 | 13 | 46.4 |
| High, 750+ | 10 | 32.3 | 11 | 39.3 |
| Smoking status | | | | |
| Current smokers | 5 | 15.6 | 7 | 22.2 |
| Non-smokers | 27 | 84.4 | 21 | 77.8 |
| Alcohol intake | | | | |
| Never | 4 | 12.5 | 6 | 21.4 |
| Sometimes | 21 | 65.6 | 16 | 57.1 |
| Frequently | 7 | 21.9 | 6 | 21.4 |
| Concentration of inorganic mercury in serum* | | | | |
| $\geq 0.2$ µg/L | 16 | 50.0 | 13 | 48.1 |
| $< 0.2$ µg/L | 16 | 50.0 | 14 | 51.9 |

*SD*: Standard deviation; *MUPS*: Medically unexplained physical symptoms. * Data available for n = 27 in the MUPS cohort.

## Estimation of unit costs

Generally speaking, Norway has universal health and social insurance coverage, where health care services are publicly financed, and policies have little direct financial implications on private health expenditures [27,28]. Data on direct and indirect health care costs were collected for all patients and analysed to inform the economic evaluation. Data on direct costs included health care costs, transport costs and the intervention costs of removing the amalgam. Indirect health care costs were the value of lost time for health care visits. Costs were determined using a micro-costing approach at the individual level. All costs were specified in Norwegian kroner (NOK), where 100 NOK is approximately equal to 12 US dollars or 10 EURO at the time of this analysis. Detailed direct and indirect costs were estimated based on primary survey data and the most recent relevant Norwegian data (S1 Table).

## Direct health care costs

Both the cost of the dental treatment (amalgam removal and replacement with other dental restorative materials) and the health care costs associated with the treatment of health problems attributed to MUPS were incorporated in the model. The latter was similar for both the intervention and the comparator and included costs of visits to general practitioners (GP),

other (private) doctors, physiotherapists, and psychologists, as well as expenses of seeking complementary and alternative medicine (CAM). Medication costs and costs of hospital admission were also included. The unit cost per GP visit was NOK 297, while for physiotherapy it was NOK 236 per visit [29]. The cost per visit of other (private) doctors was assumed to be NOK 656 [30]. The cost per visit to a psychologist was assumed to be NOK 1 500 (expert estimates). The cost of CAM was based on the survey performed by Norway's National Research Center in Complementary and Alternative Medicine in 2018 [31]. We estimated unit costs by multiplying the average annual cost per user of CAM in the population-based survey (NOK 3078) with the proportion of CAM users in the Amalgam and MUPS cohorts, respectively. Finally, the average cost per inpatient day in a medical ward of a Norwegian hospital was estimated at NOK 10 000 in 2012 prices [32,33], and NOK 11 317 after adjustment to 2018 prices.

The costs of the amalgam removal applied only for the intervention group and was equal to NOK 13 552 per person [16]. We estimated the intervention costs (amalgam restorations) based on the fees structure of the Public Dental Health Care Services.

## Transport costs

We assumed that respondents use either public transport, private car, or taxi to receive treatment or therapies. In Norway, the government refunds its employees by NOK 4.03 per kilometer for use of private cars in public service, which is estimated to cover normal levels of capital costs, maintenance, fuel and insurance. Lacking any information on the actual means of transport, we therefore used NOK 4.03 per kilometer as an estimate of the average travel cost across all types of transport. Since all patients were living in the municipalities with dental or other facilities, the one-way travel distance was set to 10 km.

## Indirect health care costs and productivity losses

We defined indirect health care costs as the value of time spent seeking health care, including consulting GPs, other (private or specialist) doctors, psychologists, and physiotherapists as well as hospital stays and dental treatment. Respondents were asked to report the number of times they had visited a health care provider during the previous three months. We assumed 2 hours' time spent per visit (including travel time, waiting time and consultation time), except for visits to psychologists, for which we assumed 3 hours as it involves longer consultation times. We estimated the value of time spent based on the average pre-tax monthly salary rate of NOK 45 610 from national statistics, which is reasonable since the study population were adults of working age. For patients who work 7.5 hours per day for 22 working days per month, the cost per lost hour was NOK 276 [i.e., NOK 45 610/(7.5*22)]. We used this information to value the days of absenteeism from work and daily compensations for sick leave. The inclusion of such productivity costs is not encouraged for economic evaluations in Norway [34], and hence costs of productivity lost were applied only in a scenario analysis.

## Health outcome measures

We used quality adjusted life years (QALYs) as a main outcome measure. The QALY combines length of life and the quality of that life into a single index, which allows for comparisons of effectiveness both within and across various conditions and groups of patients [35]. We measured the health state utilities ('Q' in QALY) using the EuroQol five-dimensional five-level (EQ-5D-5L) questionnaire, which is the most widely applied instrument in cost-effectiveness analysis [36,37]. We calculated expected QALYs by multiplying utility weights (given by EQ-5D-5L value set) with the number of life years of the assumed time horizons of the model. As a

Norwegian value set for the EQ-5D-5L is not presently available, we applied a value set derived from stated preference data of members of the English general public [38]. In the absence/advent of a national value set, the Norwegian Medicines Agency recommends UK's cross-walk value set for single technology assessments [39]. Thus, in the scenario analysis, we also applied utility weights from UK general population calculated using cross-walk algorithm; i.e., mapping the EQ-5D-5L descriptive system data onto the EQ-5D 3-level (EQ-5D-3Lvalue set) from van Hout et al [40]. We denote the latter as EQ-5D-CW to distinguish from EQ-5D-5L. We used health outcome data on the individual level responses to the five dimensions of the EQ-5D questionnaire at baseline and follow-up for both Amalgam and MUPS cohorts. A detailed description of EQ-5D-5L was given elsewhere [17].

There are only three missing values on EQ-5D-5L (one in Amalgam cohort at baseline and two in MUPS cohort at follow-up), which were replaced by mean imputation. The use of the mean of the non-missing observations is a common technique, especially in cases where the number of missing observations is quite few as the case in the present study.

### Discounting

Future costs and health outcomes were discounted at 4% per annum [41]. Scenario analyses have been done by changing the discount rate to either 0 or 6% (the recommended discount rate for measures where considerable systematic risk may reasonably be assumed).

### Decision model

We produced a novel decision model (Fig 1) to capture the costs and consequences of interventions over the time horizon of the analysis, which was 2 years at baseline and 5 and 10 years in scenario analyses. The decision model was designed to combine information about costs and outcomes, and to analyse implications of uncertain information [21].

We considered a decision tree to be an appropriate approach to accurately analyse and aggregate data that describe the cumulative effect of intervention compared with no intervention [42]. The prospective cohort study did not yield well suited information to model potential chronic and recurring properties of the condition beyond the observation period, and a Markov modelling approach was therefore not applied.

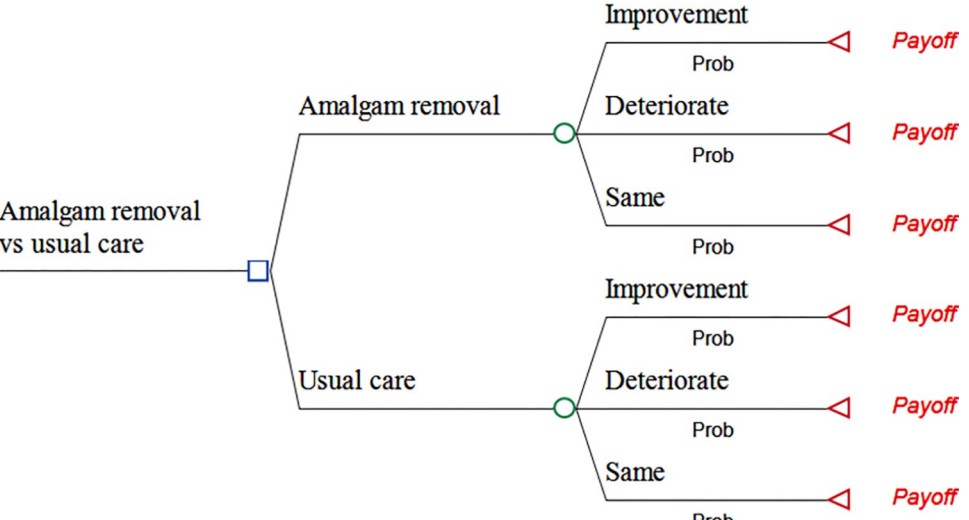

**Fig 1. Decision model tree structure showing comparison of amalgam fillings removal (AR), and usual care (UC).** The squares denote decision nodes, circles denote probability nodes, and triangles denote the terminal nodes.

We assumed that at two years following the baseline survey, patients could have converged from their initial state of deprived health into any of the three health states.

i. Improvement in health,

ii. Worsening of health, or

iii. Remain the same.

These health states were determined by considering the minimally important difference (MID) of EQ-5D-5L utilities between baseline and follow-up periods. A MID of 0.037 was considered a meaningful change [43]. Thus, "Deterioration" was defined as >0.037 reduction in health utility between baseline and follow-up, "Improvement" was defined as >0.037 increase in utility, and "The same" was classified as change in utility varying between– 0.037 and +0.037. The cohort data was then analysed to estimate the probabilities of ending up in one of the described health states (Table 2). We then estimated the mean conditional utility values for each of the three health states. This was done using the pooled sample of both the MUPS and Amalgam cohorts, assuming that the utility values for the described health states are similar for both cohorts.

To measure the incremental cost per QALY gain of Amalgam removal intervention, we used the incremental cost-effectiveness ratio (ICER) by dividing the difference in total costs between the Amalgam removal intervention and the comparison group by the difference in total QALYs. Thus, the ICER of Amalgam removal (AR) intervention versus usual care (MUPS) is defined as:

$$\frac{Costs\ AR - Costs\ MUPS}{QALYs\ AR - QALYs\ MUPS}$$

In addition to the base-case analysis with a two-year time horizon, we examined the potential cumulative incremental values (utilities and costs) over 5 and 10 years following the removal of amalgam fillings. Costs of productivity loss due to amalgam adverse symptoms were also included in a scenario analysis. Furthermore, we conducted one-way and

**Table 2. Utility and probability parameters in different health states.**

*A. Health state probabilities and utility values*

| | Health state probabilities (%) | | Health state utilities [a] |
|---|---|---|---|
| | **AR (intervention)** | **MUPS (UC)** | **Mean (SD)** |
| Deterioration (reduction >0.037) | 9.4 | 46.4 | 0.599 (0.137) |
| The same (difference of ± ≤0.037) | 21.9 | 14.3 | 0.753 (0.174) |
| Improvement (increase >0.037) | 68.7 | 39.3 | 0.784 (0.145) |

*B. Summary of EQ-5D-5L in the baseline and follow-up, mean (SD)*

| | Amalgam cohort | MUPS cohort |
|---|---|---|
| Baseline | 0.609 (0.216) | 0.696 (0.170) |
| Follow-up | 0.767 (0.157) | 0.698 (0.165) |
| Mean change | 0.158* (0.207) | 0.002 (0.146) |

*AR*: Amalgam removal; *MUPS*: Medically unexplained physical symptoms; *UC*: Usual care; *SD*: Standard deviation; *EQ-5D*: EuroQol 5-dimension.

[a] Utility values for the described health states are similar for both the intervention and the MUPS.

* p < 0.001. The data in panel A were used as model input, while the data in panel B validates the implication that amalgam removal has the potential to improve HRQoL.

probabilistic sensitivity analyses for main input parameters. The probabilistic sensitivity analyses were based on a Monte Carlo simulation with 10 000 iterations.

Although there is no officially approved cost-effectiveness threshold in Norway, a White Paper endorsed by Parliament suggests a cost-effectiveness threshold that ranges between NOK 275 000 and NOK 825 000 per QALY [44]. The proposed threshold levels vary by severity of disease, where the minimum threshold should correspond to the opportunity cost in the health services. Thus, a cost-effectiveness threshold range between NOK 275 000 and NOK 825 000 per QALY was used as a reference. The decision tree model was performed using TreeAge Pro 2020 and Excel, and the remaining analyses were conducted using Stata ver. 16.1 (StataCorp., Texas, USA).

### Ethical aspects

The study was approved by the local research ethics committee (REK2012/331) and registered at ClinicalTrials.gov (https://clinicaltrials.gov/ct2/show/NCT01682278; NCT01682278). Informed written consent was obtained from all participants in the study.

## Results

### Baseline characteristics

Baseline characteristics are summarised in Table 1. A total of 60 subjects (32 Amalgam cohort and 28 MUPS) were available at both baseline and follow-up. Mean (SD) ages were 52 (7.5) years in the amalgam cohort and 50 (10.3) years in MUPS cohort. 54% of the Amalgam cohort and 33% of the MUPS cohort, had higher than upper-secondary education. In both cohorts, the majority were female respondents. About 15.6 and 22.2% were current smokers in the amalgam and MUPS cohort, respectively. The proportion of women was lower in the Amalgam group (59.4%) than in the MUPS group (85.7%) (p = 0.042).

### Costs and utility

The probability of deterioration in health utilities was far lower in the intervention (amalgam removal) compared to usual care (MUPS), and mean health utilities were substantially lower in the deterioration group (Table 2A). For the amalgam cohort, the mean (SD) EQ-5D-5L utility increased significantly by 0.158 (0.216) between baseline and follow-up (p < 0.001). In contrast, no significant change was observed for the MUPS cohort (p = 0.970)–mean (SD) baseline EQ5D-5L for the MUPS cohort was 0.696 (0.170), while the follow-up value was 0.698 (0.165). While the health utilities by treatment arm were not directly utilised by the model, they serve to validate the implication that amalgam removal has a potential to improve quality of life (Table 2B).

Annual costs per patient are detailed in S1 Table. Mean (SD) overall intervention costs were NOK 17 630 (13 490), which is a onetime cost and not subject to discounting. The mean direct and indirect health care costs were similar for the Amalgam and MUPS cohorts at baseline, but differed at follow-up. The mean (SD) baseline direct health care costs were NOK 25 577(37 062) and NOK 23 714 (24 027) in the Amalgam and MUPS cohorts, respectively. In the follow-up period, these costs were higher in the Amalgam cohort (NOK 35 344) than in the MUPS cohort (NOK 29 166). The indirect health care costs were also similar in the MUPS and Amalgam cohorts at baseline, while at follow-up this cost was slightly higher in the MUPS cohort. Total health care costs increased between baseline and follow-up for both the amalgam and MUPS groups, but slightly (and insignificantly) more in the latter. In the Amalgam cohort,

**Table 3. Costs and utility parameters with different time horizons.**

| Input parameters | | | 95% CI | | |
|---|---|---|---|---|---|
| | **Mean** | **SE** | **Low** | **High** | **Distribution** |
| **At 2 years follow-up** | | | | | |
| *Costs (NOK)* | | | | | |
| Treatment: AR | 17 630 | 2 385 | 12 766 | 22 493 | Gamma |
| Intervention: AR | 80 400 | 16 047 | 44 397 | 92 823 | Gamma |
| Usual care: UC | 78 614 | 11 592 | 52 780 | 119 459 | Gamma |
| *Utilities (QALYs)* | | | | | |
| Deterioration | 1.152 | 0.062 | 1.028 | 1.277 | Normal |
| Same | 1.42 | 0.110 | 1.221 | 1.618 | Normal |
| Improvement | 1.486 | 0.047 | 1.392 | 1.581 | Normal |
| **At 5 years follow-up** | | | | | |
| *Costs (NOK)* | | | | | |
| Intervention: AR | 157 052 | 26 923 | 99 286 | 191 236 | Gamma |
| Usual care: UC | 206 385 | 41 431 | 120 844 | 306 937 | Gamma |
| *Utilities (QALYs)* | | | | | |
| Deterioration | 2.720 | 0.147 | 2.426 | 3.014 | Normal |
| Same | 3.352 | 0.236 | 2.883 | 3.819 | Normal |
| Improvement | 3.508 | 0.111 | 3.285 | 3.732 | Normal |
| **At 10 years follow-up** | | | | | |
| *Costs (NOK)* | | | | | |
| Intervention: AR | 266 366 | 42 434 | 177 564 | 331 588 | Gamma |
| Usual care: UC | 388 603 | 83 986 | 217 912 | 574 305 | Gamma |
| *Utilities (QALYs)* | | | | | |
| Deterioration | 4.956 | 0.268 | 4.052 | 5.034 | Normal |
| Same | 6.108 | 0.430 | 4.814 | 6.378 | Normal |
| Improvement | 6.391 | 0.203 | 5.487 | 6.233 | Normal |

*CI*: Confidence interval; *AR*: Amalgam removal; *UC*: Usual care; *SE*: Usual error; *NOK*: Norwegian kroner; *QALYs*: Quality Adjusted Life Years.

no visits to psychologists were reported in the baseline assessment, and no hospital stays in the follow-up period.

The base-case input parameters for the decision model are presented in Table 3, together with information about inference used in the subsequent sensitivity analyses (see below). The cumulative QALYs at 2 years are estimated to be higher among those who experienced improvement in health.

## Cost-effectiveness analysis

Table 4 summarises costs and effectiveness of amalgam removal compared to the usual care (MUPS cohort), including both absolute and incremental values. In the base-case model of 2 years, expected costs were higher in the amalgam group (NOK 98 030) than in the usual care group (NOK 78 614), with a difference of NOK 19 416. Under the base case assumptions, there were 1.44 QALYs in Amalgam cohort and 1.32 QALYs in the MUPS cohort, indicating an incremental effect of 0.119 QALYs for the Amalgam cohort. Thus, the incremental cost per QALY was NOK 162 680.

In the scenario analyses, when the calculation period was extended to 5 or 10 years, amalgam removal had lower expected costs than usual care while remaining better in terms of

**Table 4. Cost-effectiveness of amalgam removal in the base-case and various scenario analyses.**

| | UC: MUPS | | Intervention: AR | | Incremental | | |
|---|---|---|---|---|---|---|---|
| | Cost | QALY | Cost | QALY | ΔCost | ΔQALY | ICER |
| At 2 years follow-up (base-case) | 78 614 | 1.321 | 98 030 | 1.44 | 19 416 | 0.119 | 162 680 |
| At 5 years follow-up | 206 385 | 3.118 | 174 682 | 3.4 | -31 703 | 0.282 | Dominance¤ |
| At 10 years follow-up | 388 603 | 5.682 | 283 996 | 6.195 | -104 607 | 0.513 | Dominance |
| *EQ-5D-CW utility** | | | | | | | |
| At 2 years follow-up (base-case) | 78 614 | 1.106 | 98 030 | 1.243 | 19 416 | 0.138 | 141 198 |
| At 5 years follow-up | 206 385 | 2.61 | 174 682 | 2.935 | -31 703 | 0.324 | Dominance |
| At 10 years follow-up | 388 603 | 4.756 | 283 996 | 5.347 | -104 607 | 0.592 | Dominance |
| *Discount rate = 0%* | | | | | | | |
| At 2 years follow-up (base-case) | 78 614 | 1.401 | 98 030 | 1.528 | 19 416 | 0.126 | 153 493 |
| At 5 years follow-up | 206 385 | 3.503 | 174 682 | 3.819 | -31 703 | 0.316 | Dominance |
| At 10 years follow-up | 388 603 | 7.005 | 283 996 | 7.638 | -104 607 | 0.632 | Dominance |
| *Discount rate = 6%* | | | | | | | |
| At 2 years follow-up (base-case) | 78 614 | 1.284 | 98 030 | 1.401 | 19 416 | 0.116 | 167 161 |
| At 5 years follow-up | 206 385 | 2.951 | 174 682 | 3.217 | -31 703 | 0.266 | Dominance |
| At 10 years follow-up | 388 603 | 5.156 | 283 996 | 5.621 | -104 607 | 0.466 | Dominance |
| *With productivity costs* | | | | | | | |
| At 2 years follow-up (base-case) | 106 657 | 1.321 | 114 185 | 1.44 | 7 528 | 0.119 | 63 075 |
| At 5 years follow-up | 335 568 | 3.118 | 196 596 | 3.4 | -138 972 | 0.282 | Dominance |
| At 10 years follow-up | 662 027 | 5.682 | 314 125 | 6.195 | -347 902 | 0.513 | Dominance |

*QALY*: Quality-adjusted life year; *ICER*: Incremental cost-effectiveness ratio; *UC*: Usual Care. ¤ Dominance appears because Amalgam removal is less costly than MUPS, while at the same time being more effective. With dominance, Amalgam removal is cost-effective irrespective of willingness to pay for health.

health effects. Thus, amalgam removal became cost-saving and the dominant intervention (Table 4). In the scenarios with dominance, amalgam removal may be considered cost-effective irrespective of the level of willingness to pay for health. When utilities were measured using the cross-walk algorithm (EQ-5D-CW), the incremental QALYs increased slightly and the ICER with a 2-year time horizon further improved to 141 198 NOK per QALY. We also conducted a scenario analysis by including the value of lost productivity. Under this condition, the incremental cost per QALY substantially decreased from NOK 162 680 in the base case to NOK 63 075 per QALY, favouring amalgam removal. Cost-effectiveness results were robust to changes in the discount rate, which happened because costs and utilities were modelled largely as concurrent streams of values.

One-way sensitivity analyses for major input variables were depicted by Tornado diagram (Fig 2). The most important variables were the health care costs in both Amalgam and MUPS cohorts. The lower (higher) the costs of the usual care (amalgam removal), the higher the ICER was for the amalgam removal intervention. However, these changes in health care costs led to ICER estimates that remained within levels that are usually considered as cost-effective in Norway. The ICER was also moderately sensitive to the utility and probability of health deteriorations in the MUPS cohort.

Finally, the results of the probabilistic sensitivity analysis with 10 000 iterations are presented in Fig 3. The *upper panel* depicts the cost-effectiveness scatter plot, and illustrates that a majority of the cost-effectiveness pairs fall below and to the right of the minimum cost-effectiveness threshold line (with slope of NOK 275 000), indicating a high probability that the intervention is cost-effective. The probability that the amalgam removal would be cost-effective compared to usual care at different thresholds of willingness to pay is depicted in the cost-

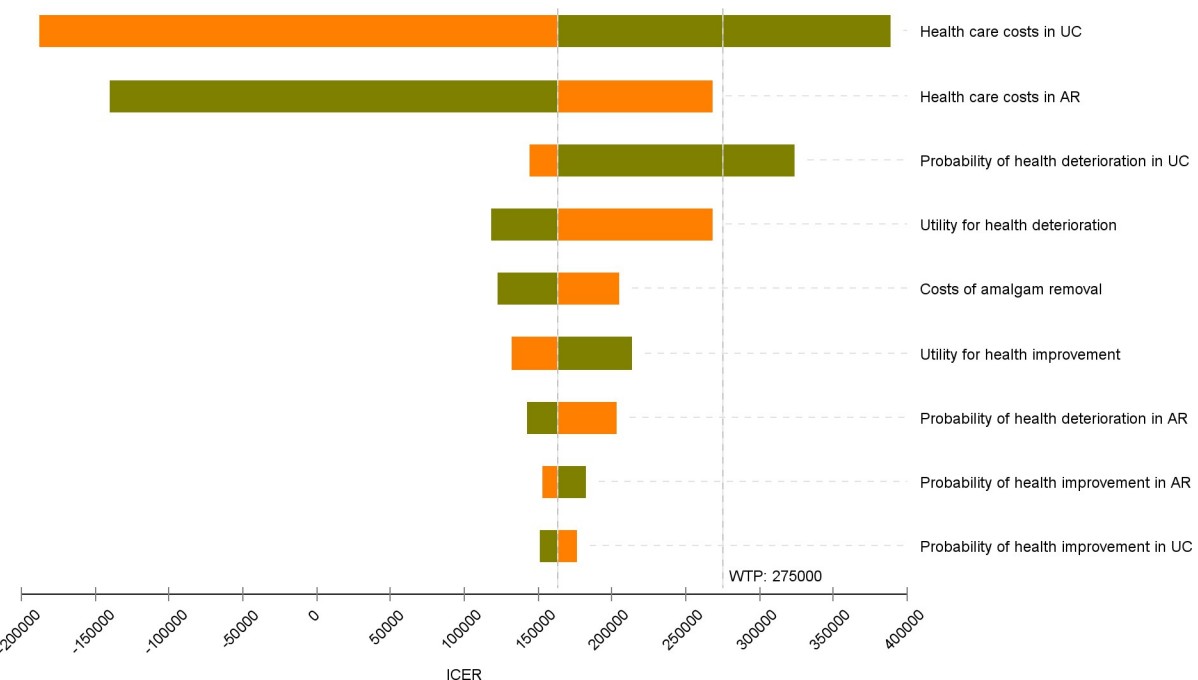

**Fig 2. Tornado diagram illustrating results from one-way sensitivity analyses.** *UC*: Usual care; *AR*: Amalgam removal; *ICER*: Incremental cost-effectiveness ratio.

effectiveness acceptability curve in the *bottom panel* of Fig 3. The cost-effectiveness acceptability curve shows that when willingness to pay is higher than NOK 165 372 per QALY, amalgam removal has a higher probability of being cost-effective than usual care. At a willingness to pay of NOK 275 000, there was over 72% probability that amalgam removal is cost-effective and this probability increased further with higher levels of willingness to pay.

## Discussion

This study investigated the cost-effectiveness of the removal of amalgam restorations in patients with MUPS that they attribute to amalgam fillings (the Amalgam cohort), compared to the usual care (i.e., patients *without attribution* to dental amalgam restorations and no amalgam removal–the MUPS cohort). Based on our cost-effectiveness analyses, removal of amalgam restorations is more costly, but more effective than the usual care at base case assessment, yielding an incremental cost of NOK 162 680 per QALY. Amalgam removal is thus likely to be considered cost-effective in Norway, in the light of a commonly assumed threshold of NOK 275 000 per QALY. In general, evaluating both the effectiveness and costs of the interventions provides information that is important not only to the providers and the recipients of the treatment, but also to those planning treatment programmes or financing the reimbursement for the procedure.

We have not been able to identify any published cost-effectiveness analyses of amalgam removal compared to usual care in patients with MUPS attributing their symptoms to amalgam fillings. Existing economic studies in restorative dentistry compare different forms of dental materials in restorative therapy. They generally find that amalgam fillings are cost-effective compared to their alternatives in terms of durability [24,45,46].

Costs and health outcomes are important to consider when comparing alternative treatments or interventions. Obviously, the ideal treatment alternative is the one that has both the

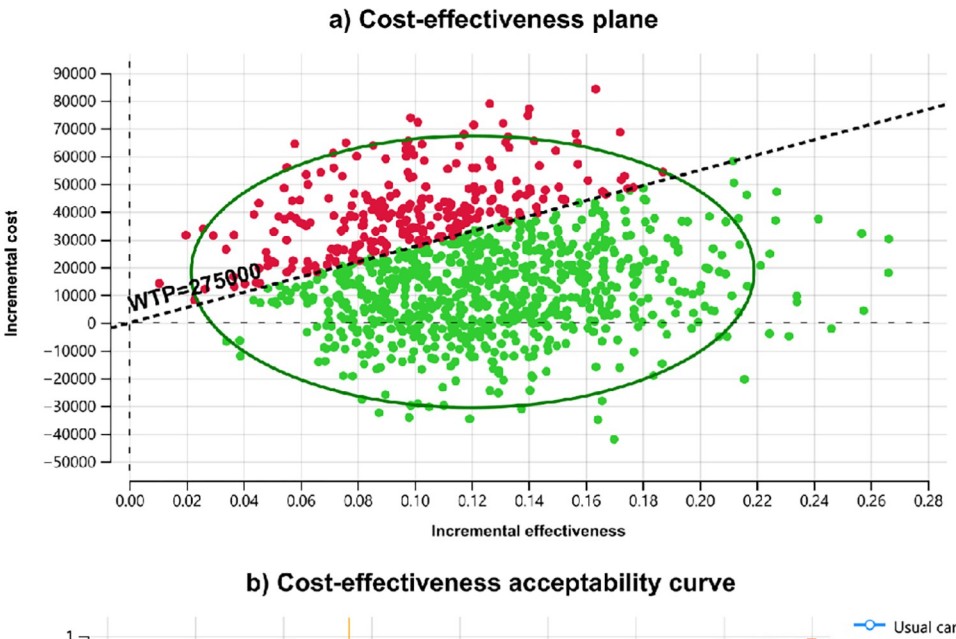

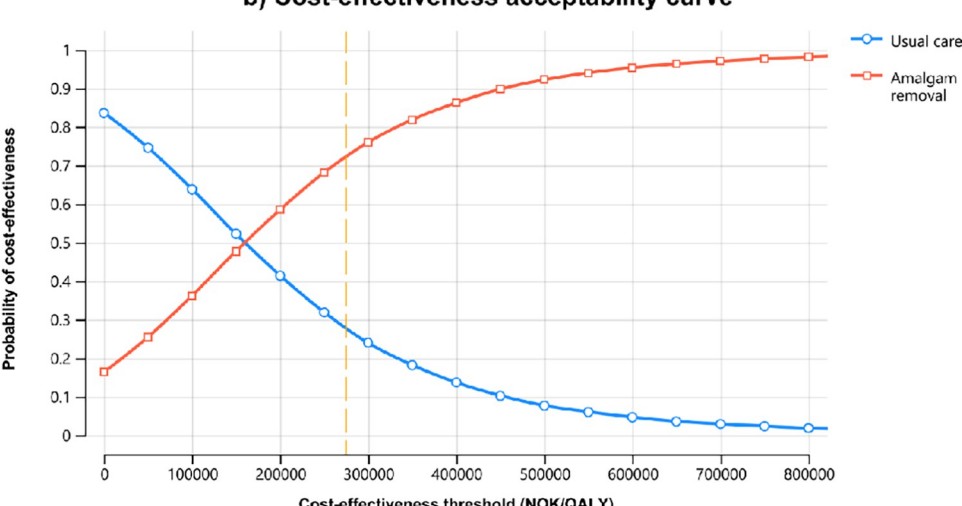

**Fig 3. Results from Monte Carlo simulation with 10 000 iterations.** Panel a) illustrates the cost-effectiveness pairs for 'Amalgam removal' compared to 'Usual care', and panel b) shows the cost-effectiveness acceptability curves that illustrate the probabilities that either intervention alternative is cost-effective for a range of cost-effectiveness thresholds. Dashed (orange) line is the minimum cost-effectiveness threshold of 275 000 NOK.

lowest cost and the highest effectiveness. In our base case analysis, the removal of amalgam restorations had higher effectiveness, but at a higher cost like most other cost-effectiveness studies. When cost-effectiveness calculations were based on 5 and 10-year time horizons, amalgam removal had lower costs as well as higher effectiveness compared to the usual care, resulting in dominant strategies in both scenarios. This conclusion is based on a societal perspective that takes into account the combined costs of patients and service providers, and depends on the assumption that the improved health outcomes and reduced direct and indirect health care costs that we observed for the amalgam group at 2 years will be sustained in the longer run. A longer follow-up survey will provide more information about the realism of these scenarios.

Another important point of discussion is whether to incorporate productivity costs into the analysis, or the value of reduced production due to ill-health. A review of literature on cost-

effectiveness analyses showed a considerable variability in costing methods [47]. Some suggest that only losses due to time spent seeking and obtaining care should be included, while productivity losses caused, for example, by sick leave should not be included [48]. Others argue that the time spent seeking care does not result in any additional productivity loss when patients are already off work due to their illness, and hence this cost is not relevant [21]. We did not include the value of loss of production due to ill-health in our base case analysis, based on Norwegian guidelines that discourage the inclusion of such loss in cost-effectiveness analysis [34]. The inclusion of productivity costs in scenario analyses further strengthened our conclusion of amalgam restoration being cost-effective.

We also assessed how sensitive the results were to uncertainties in major input parameters. The ICER was most sensitive to uncertain costs both for the intervention and usual care. This is not surprising since these parameters had wide confidence intervals. A decrease in total health care costs for usual care by 20% increased the incremental cost per QALY by over 40% compared to the base-case, potentially making amalgam removal not cost-effective with a willingness to pay-off 275 000 NOK per QALY. A decrease in the probability of health deterioration with usual care was the only other parameter that could potentially make the intervention not cost-effective.

Under the scenario analysis with EQ-5D-CW as a health outcome measure, the ICER of the intervention becomes more favourable. This is because the EQ-5D-CW produce slightly higher utility scores compared to the EQ-5D-5L, and hence improvements in quality of life are valued higher with the EQ-5D-CW than when the EQ-5D-5L is used. This mean difference is mainly attributed to the effect of the change in valuation system. Though the two measures have the same descriptive system, the EQ-5D-CW uses the EQ-5D-3L utility values. Other evidence also suggests that the ICER of an intervention becomes higher (or becomes less cost-effective) if the EQ-5D-5L instrument is used in place of the EQ-5D-3L instrument [49].

A strength of this study is the comprehensive coverage of health care costs, including such parameters as patient time, patients' value of time, travel expenses, and patients' value of different health states. This is valuable information for planning and priority setting. We also conducted several scenario and sensitivity analyses to check the consistency of our base-case analysis. The results show that our conclusion remains unchanged after relaxing the values of key parameters. The model is generally robust to wide variations in other parameters.

This study has several limitations. While medication costs and costs for hospital stays were major cost components, we have no means of firmly attributing them to health problems caused by amalgam restorations, and hence our results are potentially biased by health problems unrelated to MUPS. To account for this uncertainty, we varied costs over wide ranges, and found our results to be only modestly affected. The study was not randomised, and the two groups were non-equivalent regarding symptom attribution to amalgam fillings (100% vs. 0% of subjects in the amalgam and MUPS group, respectively) as well as recruitment setting [16]. Thus, selection bias from differences in prognostic factors that were not documented or are unknown (as well as a difference in gender distribution) is possible, which limits the interpretation of the results. On the other hand, the two patient groups fulfilled identical criteria for MUPS and had a similar symptom load at baseline.

Specifically, if recovery from a chronic disease in the Amalgam cohort is more frequent than a similar recovery in the MUPS cohort, the results could be disproportionally biased in favour of amalgam restoration. However, all patients were examined by their GP before inclusion [16] and all diseases were adequately treated before inclusion. Thus, we believe the risk for bias from this is minimised within the limitations of the study design. Also, the comparison with the MUPS cohort will control for effects related to regression to the mean and natural improvement. Inclusion of failure rates of the dental restorations would have made our cost-

effectiveness analysis more sound. Thus, lack of information on the annual failure rate for the restorations placed in this study could be considered as a limitation.

Finally, given the small sample size and several inclusion and exclusion criteria, generalisation to the broader population that might be covered in case of public funding of the intervention must be done with caution. Future studies with larger sample size and, if possible, a randomised design, are desirable. However, randomised clinical trials on the effects from removal of amalgam fillings in patients with health complaints attributed to their amalgam fillings are associated with methodological challenges, including patient preferences for amalgam removal and associated expectations with potential placebo effects [50]. However, in this study, possible side effects and complications associated with the amalgam removal (e.g., need for endodontic treatment or revision of placed restorations due to technical complications) are trivial within the follow up period. While potential small health effects may be assumed to be captured by the QALY estimates, any costs of side effects and complications were not considered.

In summary, this study provides understanding about the costs and health outcomes associated with the removal of amalgam restorations in patients who attribute health complaints to dental amalgam restorations and demonstrate that the intervention is likely to be cost-effective compared to usual care. Although the results from the study are subject to sample size limitations and possible biases from the non-randomised design, they are based on real programme experience and provide plausible evidence of beneficial effects from removing dental amalgam in both short- and long-term perspectives in patients who attribute health complaints to dental amalgam restorations and fulfil the used inclusion and exclusion criteria. Although there is no concrete evidence that amalgam causes general health symptoms, the beneficial effects from removal of amalgam restorations are consistent with several previous studies [13,14,51,52].

## Supporting information

**S1 Table. Annual per person costs in the Amalgam and MUPS cohort (in NOK) at baseline and follow-up.**
(DOCX)

## Acknowledgments

We thank Merete Allertsen and Randi Sundfjord for excellent help with the project administration and technical assistance.

## Author Contributions

**Conceptualization:** Admassu N. Lamu, Lars Björkman, Bjarne Robberstad.

**Data curation:** Admassu N. Lamu, Lars Björkman, Bjarne Robberstad.

**Formal analysis:** Admassu N. Lamu, Bjarne Robberstad.

**Funding acquisition:** Lars Björkman.

**Investigation:** Admassu N. Lamu, Lars Björkman, Bjarne Robberstad.

**Methodology:** Admassu N. Lamu, Bjarne Robberstad.

**Project administration:** Lars Björkman.

**Software:** Admassu N. Lamu.

**Supervision:** Lars Björkman, Bjarne Robberstad.

**Validation:** Admassu N. Lamu, Harald J. Hamre, Terje Alræk, Frauke Musial, Bjarne Robberstad.

**Writing – original draft:** Admassu N. Lamu.

**Writing – review & editing:** Admassu N. Lamu, Lars Björkman, Harald J. Hamre, Terje Alræk, Frauke Musial, Bjarne Robberstad.

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
