## [Decision Letter · Decision Letter 0]

14 Feb 2022

PONE-D-22-00123Is amalgam removal in patients with medically unexplained physical symptoms cost-effective? A prospective cohort study in NorwayPLOS ONE

Dear Dr. Lamu,

Thank you for submitting your manuscript to PLOS ONE. After careful consideration, we feel that it has merit but does not fully meet PLOS ONE’s publication criteria as it currently stands. Therefore, we invite you to submit a revised version of the manuscript that addresses the points raised during the review process.

We look forward to receiving your revised manuscript.

Kind regards,

Kelvin Ian Afrashtehfar, M.Sc., D.D.S.,Dr. med. dent., FRCDC

Academic Editor

PLOS ONE

Journal Requirements:

Additional Editor Comments:

Dear Authors,

After appraising the manuscript and the unanimous decision of the reviewers, I concede a major revision to it.

Please make use of the reviewers comments/

You can mention in the introduction that amalgam is still considered a treatment option. The supporting literature follows:

- Int Endod J. 2017 Oct;50(10):951-966. doi: 10.1111/iej.12723. Epub 2017 Jan 17.

- J Prosthet Dent. 2017 Mar;117(3):345-353.e8. doi: 10.1016/j.prosdent.2016.08.003.

Best Regards,

Reviewers' comments:

Reviewer's Responses to Questions

**Comments to the Author**

1. Is the manuscript technically sound, and do the data support the conclusions?

Reviewer #1: Partly

Reviewer #2: Yes

Reviewer #3: Yes

Reviewer #4: Yes

2. Has the statistical analysis been performed appropriately and rigorously? 

Reviewer #1: Yes

Reviewer #2: I Don't Know

Reviewer #3: Yes

Reviewer #4: Yes

3. Have the authors made all data underlying the findings in their manuscript fully available?

Reviewer #1: Yes

Reviewer #2: Yes

Reviewer #3: Yes

Reviewer #4: Yes

4. Is the manuscript presented in an intelligible fashion and written in standard English?

Reviewer #1: No

Reviewer #2: Yes

Reviewer #3: Yes

Reviewer #4: Yes

5. Review Comments to the Author

Reviewer #1: 1. The whole manuscript needs English copy editing

2. Introduction section. The author needs to highlight more on the main problem of the study. It's a bit ambiguous here, whether the main problem is MUPS or the high demand of amalgam replacement. Please kindly explain

3. Method section.

a. The study was performed excellently, with so many details, which is good.  Please add more explanation about the formulating calculation that were used to analyze those acquired information.

b. Please explain why amalgam replacement need 12 months to be done. Was that the required time to replace 1 restoration or all the restorations in 1 patient.

4. The result section. Please move all the tables to the result section. I think it would be better if all the variables appear in the result are tabulated (ex: please write both male and female on table 1)

5. Discussion section. The age of the samples varies between 20-70 years old. Please add an explanation whether there are any confounding factors, like the respondents general health, thay might affect the result.

Reviewer #2: What is standard care that you refer to and compare to?

Can you briefly explain how the health care system works in the country in terms of financial burden on individual?

I think that the interpretation of what WHO said about dental amalgam is different from what WHO said. The danger is mostly from placent and removal and affects dentist and dental staff more thatn the patient. WHO says that there is lack of evidence to danger of dental amlgam,

Is there a relationship between number of removed amalgams and improvement?

TABLE 1 does not include male information under gender

Did you perform a power calculation to determine sample size?

What is the average number of amalgam fillings per patient in the amalgam group? And how is that corelated with study outcomes?

What is the average age of amalgam in each patient? What are the conditions of amalgams prior to removal? Looking at failure rate of fillings/crowns that replaced those amalgam would give a complete picture or cost-effectiveness of removal. I understand that the data is not available now but could be considered as a limitation.

Reviewer #3: All is appropriately done. However, page 3, there is need to correct the references 5 and 6 in line with the journal requirements

Well written paper. Correct references 5 and 6 on page 3 to conform with journal requirements

Reviewer #4: This study titled ‘’ Is amalgam removal in patients with medically unexplained physical symptoms cost effective? A prThis study titled ‘’ Is amalgam removal in patients with medically unexplained physical symptoms cost effective? A prospective cohort study in Norway” assessed the cost-effectiveness of the removal of amalgam restorations in patients with medically unexplained physical symptoms (MUPS) attributed to amalgam fillings compared to standard care, based on a prospective cohort study in Norway. The study provides understanding about the costs and health outcomes associated with the removal of amalgam restorations in patients who attribute health complaints to dental amalgam restorations. The study emphasized that the available clinical and cost data indicate that amalgam fillings removal in patients who attribute health complaints to dental amalgam restorations is a highly cost-effective intervention.

It also highlighted that the estimated incremental cost per QALY decreased with increasing time horizon, and amalgam removal was found to be cost saving over both 5 and 10 years. The study is indeed interesting, and it tackles an important topic for both dentists and policy makers. Overall, the paper was very well written, but it needed some revisions to improve the clarity. In this context, some clarifications are needed.

Methods

1. The reason why you recruited a healthy group (Healthy cohort) in your study is not clear, your analysis and interpretation of data focused primarily on the amalgam cohort and the MUPS cohort. Accordingly, there is no need to mention the third group that was not included in your analysis.

2. You mentioned that the inclusion and exclusion criteria for the subjects are published in another article (15). However, some of these criteria must be included in your study e.g whether the MUPS group had amalgam fillings or not? , and the minimal number of fillings in the amalgam cohort.

3. The sample size calculation was not mentioned and accordingly it is not clear why you chose to include 60 patients in your study.

Although the study has several limitations yet, it is unique in being the first study to perform a classic economic evaluation of dental amalgam fillings removal for health reasons.

6. PLOS authors have the option to publish the peer review history of their article (what does this mean?). If published, this will include your full peer review and any attached files.

Reviewer #1: No

Reviewer #2: No

Reviewer #3: **Yes: **Prof.Midion Mapfumo Chidzonga

Reviewer #4: No

---

## [Author Response · Author response to Decision Letter 0]

19 Mar 2022

Dear Editor(s),

We are most grateful for your reviews and helpful comments of our manuscript “Is amalgam removal in patients with medically unexplained physical symptoms cost-effective? A prospective cohort and decision modelling study in Norway”. We have now made careful revisions following yours and reviewers' comments. A point-by-point response to each comment is provided as a separe attachment during this online resubmission. Below is a copy of the attached response:

Point-by-point responses to comments:

(A) Journal Requirements:

Thank for reminding us. We have now considered all journal requirements. 

Thank for spotting this problem. It has now been revised as:

“Funding: The project was funded by Norwegian Ministry of Health and Care Services (Helse- og Omsorgsdepartementet) via the Norwegian Directorate of Health. The funder had no role in study design, data collection and analysis, decision to publish, or preparation of the manuscript.”

This project has no grant number or reference number. This has been reported when we submitted the manuscript.

We included all necessary data related to our analyses reported in the manuscript. However, individual data points were not included due privacy issue, and justification has been provided. 

Data Availability: Data contain potentially identifying and sensitive patient information and is not available due to personal data protection regulations. The restrictions are imposed by NORCE's Administrative Support for Research and the Data Protection Office at NORCE. Data requests may be sent to NORCE's Administrative Support for Research (forskningsstotte@norceresearch.no) with reference to "Project 42564 Prospektiv kohortstudie av helseplager."

Thank for this comment. We have now included the ethics statement in the ‘Methods’ section. It reads as follow:

“The study was approved by the local research ethics committee (REK2012/331) and registered at ClinicalTrials.gov (https://clinicaltrials.gov/ct2/show/NCT01682278; NCT01682278). Informed written consent was obtained from all participants in the study.”

(B) Additional Editor Comments:

Dear Authors,

After appraising the manuscript and the unanimous decision of the reviewers, I concede a major revision to it.

Please make use of the reviewers comments/

You can mention in the introduction that amalgam is still considered a treatment option. The supporting literature follows:

- Int Endod J. 2017 Oct;50(10):951-966. doi: 10.1111/iej.12723. Epub 2017 Jan 17.

- J Prosthet Dent. 2017 Mar;117(3):345-353.e8. doi: 10.1016/j.prosdent.2016.08.003.

Thank you for this comment. We have now included a statement including the supporting literature as per your suggestion. (Page 3, end of last par.)

(C) Reviewers' comments to the Author

Reviewer #1: 

1. The whole manuscript needs English copy editing- 

We have had another round of professional language editing of the manuscript, and hope the English language is now acceptable. 

2. Introduction section. The author needs to highlight more on the main problem of the study. It's a bit ambiguous here, whether the main problem is MUPS or the high demand of amalgam replacement. Please kindly explain 

Thank you for this comment. We have now briefly highlighted the main problem of the study (page 4, paragraph 2)

3. Method section.

a. The study was performed excellently, with so many details, which is good. Please add more explanation about the formulating calculation that were used to analyze those acquired information. 

Thank for the comment. We now included further details about the formulating calculations where necessary. 

b. Please explain why amalgam replacement need 12 months to be done. Was that the required time to replace 1 restoration or all the restorations in 1 patient. 

Thank for this comment. Mean length of the treatment period (from acceptance to start treatment to finished treatment) was 253 days (SD 134, range from 29 to 619) for all respondents in Amalgam cohort. The more amalgam surfaces to replace, the longer was the treatment period. The correlation between number of amalgam surfaces to replace and treatment period was significant (p=0.019). We have now added an explanation (Page 5, last paragraph).

4. The result section. Please move all the tables to the result section. I think it would be better if all the variables appear in the result are tabulated (ex: please write both male and female on table 1) 

We agree with the reviewer. We moved relevant tables to the RESULTS section. We also included all relevant variables in Table 1. 

5. Discussion section. The age of the samples varies between 20-70 years old. Please add an explanation whether there are any confounding factors, like the respondents general health, they might affect the result. 

Thank you for bringing the issue of confounding to our attention. We have now added explanation involving any general health conditions of the respondents (Page 20, last paragraph) 

Reviewer #2: 

What is standard care that you refer to and compare to? 

Thank you for this comment. The term “standard of care” may be confusing as it implies a uniform or proven practice standard. Thus, for MUPS patients without attribution to amalgam fillings, we replace it by the term “usual care” to describe ‘de facto’ clinical care without any value judgment. 

Can you briefly explain how the health care system works in the country in terms of financial burden on individual? 

Norway has universal health and social insurance coverage, known as the National Insurance Scheme (NIS). Health coverage is automatic for all residents and has two main funding sources: the general tax system and household out-of-pocket payments. The share of the latter is quite small, where public sources account for most health expenditures in Norway, at 85 percent. The establishment of universal coverage has a long history in Norway, and beyond the scope of this manuscript. However, we agree with the reviewer and included brief explanation (Page 7, first paragraph)

I think that the interpretation of what WHO said about dental amalgam is different from what WHO said. The danger is mostly from placent and removal and affects dentist and dental staff more thatn the patient. WHO says that there is lack of evidence to danger of dental amlgam, 

The WHO Consensus Statement on Dental Amalgam from 1997 cited by the World Dental Federation at their web pages (https://www.fdiworlddental.org/) is obsolete, and more recent research express that the thresholds for toxicity related to mercury exposures are uncertain. For example, the recent statement issued by US FDA (September 24, 2020) says “Current estimates of continuous, exposure to mercury from dental amalgam and other sources over a lifetime that are likely to be without risks of harmful effects in the general population and greater risk groups, vary considerably (https://www.fda.gov/medical-devices/safety-communications/recommendations-about-use-dental-amalgam-certain-high-risk-populations-fda-safety-communication) … these uncertainties present challenges with regard to defining a specific threshold of toxicity for chronic, low-level mercury exposure from dental amalgam and other sources, particularly for sensitive groups.” Yet, we agree with reviewer and, hence we have now revised the sentence (Page 3, para. 1)

Is there a relationship between number of removed amalgams and improvement? 

There is a positive relationship between number of removed amalgams and improvement, which will be reported elsewhere. However, a positive (but not statistically significant) correlation between number of amalgam surfaces and symptom score has been reported previously (see Weidenhammer et al 2010). The correlation between inorganic mercury in plasma (a biomarker for exposure to dental amalgam) and symptom score was significant (see Weidenhammer et al 2010, Figure 3). 

Reference:

 Weidenhammer W, Bornschein S, Zilker T, Eyer F, Melchart D, Hausteiner C. Predictors of treatment outcomes after removal of amalgam fillings: associations between subjective symptoms, psychometric variables and mercury levels. Community Dent Oral Epidemiol. 2010;38:180-9.

TABLE 1 does not include male information under gender 

Thank you for spotting this issue. We have now revised.

Did you perform a power calculation to determine sample size? 

Thank for the comment. However, issues related to power calculation was reported elsewhere (Björkman et al., 2020), entitled “Removal of dental amalgam restorations in patients with health complaints attributed to amalgam: A prospective cohort study.” https://pubmed.ncbi.nlm.nih.gov/32810306/ . We have now briefly indicated this in the manuscript (Page 5; Para. 1).

What is the average number of amalgam fillings per patient in the amalgam group? 

Mean number of amalgam surfaces was 20.3 (SD 10.9, range from 5 to 59). This has been reported on page 5, par. 1.

And how is that corelated with study outcomes? 

Number of amalgam surfaces at baseline was not a statistically significant predictor of change. An increase by 10 amalgam surfaces increased odds ratio only slightly (1.245; 95% CI 0.635 to 2.441; p=0.524, n=32) to have a change score of the General Health Complaints index (GHC-index) above the median value of 10.95.

Patients with higher concentration of inorganic mercury in serum at baseline had higher odds ratio to have a change score of the GHC index above the median value of 10.95. An increase of 0.5 µg Hg/L increased odds ratio by 2.161 (95% CI 0.903 to 5.174; n=32) (Björkman et al; manuscript in preparation).

What is the average age of amalgam in each patient? What are the conditions of amalgams prior to removal? Looking at failure rate of fillings/crowns that replaced those amalgam would give a complete picture or cost-effectiveness of removal. I understand that the data is not available now but could be considered as a limitation. 

Thank for raising this important point. Since the annual failure rate for the restorations placed in this study is not available, we have now acknowledged the limitations in the discussion section (Page 21, para. 1).

Reviewer #3: 

All is appropriately done. However, page 3, there is need to correct the references 5 and 6 in line with the journal requirements

Well written paper. Correct references 5 and 6 on page 3 to conform with journal requirements

Thank for your positive comments and spotting the errors. These references are now corrected.

Reviewer #4: 

This study titled ‘’ Is amalgam removal in patients with medically unexplained physical symptoms cost effective? A prThis study titled ‘’ Is amalgam removal in patients with medically unexplained physical symptoms cost effective? A prospective cohort study in Norway” assessed the cost-effectiveness of the removal of amalgam restorations in patients with medically unexplained physical symptoms (MUPS) attributed to amalgam fillings compared to standard care, based on a prospective cohort study in Norway. The study provides understanding about the costs and health outcomes associated with the removal of amalgam restorations in patients who attribute health complaints to dental amalgam restorations. The study emphasized that the available clinical and cost data indicate that amalgam fillings removal in patients who attribute health complaints to dental amalgam restorations is a highly cost-effective intervention.

It also highlighted that the estimated incremental cost per QALY decreased with increasing time horizon, and amalgam removal was found to be cost saving over both 5 and 10 years. The study is indeed interesting, and it tackles an important topic for both dentists and policy makers. Overall, the paper was very well written, but it needed some revisions to improve the clarity. In this context, some clarifications are needed.

Methods

1. The reason why you recruited a healthy group (Healthy cohort) in your study is not clear, your analysis and interpretation of data focused primarily on the amalgam cohort and the MUPS cohort. Accordingly, there is no need to mention the third group that was not included in your analysis. 

We generally agree with the reviewer that the “Healthy Cohort” was not included in our analysis in this manuscript. However, this group was part of the mother project and worth mentioning for the purpose of transparency. We have written (page 4) “This analysis is based on the Amalgam cohort (intervention group) and MUPS cohort (comparator group) and included participants who responded to both the baseline and follow-up questionnaires.” We have now included additional information directly related to the ‘healthy cohort’: “Data for the healthy cohort was not utilized, as this group is not considered eligible for the intervention” (page 5, para. 2).

2. You mentioned that the inclusion and exclusion criteria for the subjects are published in another article (15). However, some of these criteria must be included in your study e.g whether the MUPS group had amalgam fillings or not? , and the minimal number of fillings in the amalgam cohort. 

Thank for the comments. The MUPS cohort was not examined by a dentist, and thus it was not an inclusion criterion for the MUPS cohort to have amalgam fillings. However, inorganic mercury in serum (which correlates with number of amalgam surfaces) was not significantly different from the amalgam cohort (48.1 % in the MUPS cohort had values ≥ 0.2 µg/L, while 50 % in the Amalgam cohort had values ≥ 0.2 µg/L, p=0.887). Thus, there is no evidence that the exposure was different in the two cohorts. Yet, we agreed with the reviewer, and key relevant criteria have now been included (page 5, para. 1).

3. The sample size calculation was not mentioned and accordingly it is not clear why you chose to include 60 patients in your study. 

Thank for the comment. Issues related to power calculation was reported elsewhere (Björkman et al., 2020), entitled “Removal of dental amalgam restorations in patients with health complaints attributed to amalgam: A prospective cohort study.” https://pubmed.ncbi.nlm.nih.gov/32810306/

Moreover, studies strongly warn not to perform a post-hoc power analysis, because post hoc power analysis identifies population-level parameters with sample-specific statistics, makes no conceptual sense and can be misleading (Zhang et al., 2019; Lakens, 2022). Although power analysis is a crucial for planning clinical research studies, it is both conceptually flawed and analytically misleading when used to indicate power for outcomes already observed (like the present study). Despite this challenge, a previous study showed a mean difference in a general health complaint of 10.0 between groups (treatment and reference groups) with a common within standard deviation of 10.0 produced a sample size of 20 in each group with a power of 87% (a two-tailed test at α =5%) Sjursen et al., 2011). A sample size calculation was performed during planning of the mother project on which the present study is based. The calculation was reported elsewhere (Björkman et al., 2020). 

References:

1. Zhang et al. (2019). Post hoc power analysis: is it an informative and meaningful analysis? General Psychiatry;32:e100069. doi:10.1136/ gpsych-2019-100069.

2. D. Lakens (2022). A relevant blog post on: “What to do if Your Editor Asks for Post-hoc Power?” available at http://daniellakens.blogspot.com/2014/12/observed-power-and-what-to-do-if-your.html, which is also included in a paper “Sample Size Justification. Collabra: Psychology»available at PsyArXiv (https://psyarxiv.com/9d3yf/)

3. Sjursen et al. (2011). Changes in health complaints after removal of amalgam Fillings. Journal of Oral Rehabilitation; 38; 835–848.

4. Björkman et al. (2020). Removal of dental amalgam restorations in patients with health complaints attributed to amalgam: A prospective cohort study. J Oral Rehabil.;47:1422–1434. DOI: 10.1111/joor.13080.

---

## [Decision Letter · Decision Letter 1]

5 Apr 2022

Is amalgam removal in patients with medically unexplained physical symptoms cost-effective? A prospective cohort and decision modelling study in Norway

PONE-D-22-00123R1

Dear Dr. Lamu,

We’re pleased to inform you that your manuscript has been judged scientifically suitable for publication and will be formally accepted for publication once it meets all outstanding technical requirements.

Kind regards,

Kelvin I. Afrashtehfar, M.Sc., D.D.S.,Dr. med. dent., FRCDC

Academic Editor

PLOS ONE

Additional Editor Comments (optional):

Dear Authors,

Please add the formula used to calculate cost analysis.

Only then, your manuscript can be accepted without further comments.

Thank you,

Academic Editor

Reviewers' comments:

Reviewer's Responses to Questions

**Comments to the Author**

1. If the authors have adequately addressed your comments raised in a previous round of review and you feel that this manuscript is now acceptable for publication, you may indicate that here to bypass the “Comments to the Author” section, enter your conflict of interest statement in the “Confidential to Editor” section, and submit your "Accept" recommendation.

Reviewer #1: (No Response)

Reviewer #2: All comments have been addressed

Reviewer #4: All comments have been addressed

2. Is the manuscript technically sound, and do the data support the conclusions?

Reviewer #1: Yes

Reviewer #2: Yes

Reviewer #4: Yes

3. Has the statistical analysis been performed appropriately and rigorously? 

Reviewer #1: Yes

Reviewer #2: I Don't Know

Reviewer #4: Yes

4. Have the authors made all data underlying the findings in their manuscript fully available?

Reviewer #1: Yes

Reviewer #2: Yes

Reviewer #4: Yes

5. Is the manuscript presented in an intelligible fashion and written in standard English?

Reviewer #1: Yes

Reviewer #2: Yes

Reviewer #4: Yes

6. Review Comments to the Author

Reviewer #1: Thank you for the excellent response. Would you please add the formula used to calculate cost analysis?

Reviewer #2: I believe that my questions have been answered and incorporated within manuscript when appropriate .

Reviewer #4: Thank you for doing the required corrections. you have adequately addressed the comments raised. The manuscript is now a

technically sound piece of scientific research with data that supports the conclusions.

7. PLOS authors have the option to publish the peer review history of their article (what does this mean?). If published, this will include your full peer review and any attached files.

Reviewer #1: No

Reviewer #2: No

Reviewer #4: **Yes: **Amira Badran

---

## [Editor Report · Acceptance letter]

22 Apr 2022

PONE-D-22-00123R1 

Is amalgam removal in patients with medically unexplained physical symptoms cost-effective? A prospective cohort and decision modelling study in Norway 

Dear Dr. Lamu:

I'm pleased to inform you that your manuscript has been deemed suitable for publication in PLOS ONE. Congratulations! Your manuscript is now with our production department. 

Kind regards, 

on behalf of

Dr. Kelvin I. Afrashtehfar 

Academic Editor

PLOS ONE